# MODEL-BASED ROBUST ADAPTIVE SEMANTIC SEGMENTATION

## ABSTRACT

Semantic image segmentation enjoys a wide range of applications such as autonomous vehicles and medical imaging while it is typically accomplished by deep neural networks (DNNs). Nevertheless, DNNs are known to be fragile to input perturbations that are adversarially crafted or occur due to natural variations, such as changes in weather or lighting conditions. This issue of lack of robustness prevents the application of learning-based semantic segmentation methods on safety-critical applications. To mitigate this challenge, in this paper, we propose a new robust training algorithm, called MRTAdapt, for Model-based Robust Adaptive training, to enhance the robustness of DNN-based semantic segmentation methods against natural variations that leverages model-based robust training algorithms and generative adversarial networks. Natural variation effects are minimized from both image and label sides. We provide extensive experimental results on both real-world and synthetic datasets demonstrating that the proposed training algorithm result in robust models that outperform multiple state-of-the-art models under various natural variations.

## 1 INTRODUCTION

In recent years, computer vision has become one of the most promising research areas in deep learning because it has empowered a great amount of industry-level applications. In the context of computer vision, semantic segmentation is a core task formulated as a dense labeling problem Luc et al. (2016), targeting to allocate every pixel with a label Hsin et al. (2019) of what is being represented. There are many applications now being used have made great progress with the help of semantic segmentation, such as medical image processing Xue et al. (2018), autonomous vehicles Zhao et al. (2018) and robotics Wang et al. (2019). However, many applications of semantic segmentation are life-critical, which means that low model accuracy may pose direct threat to human safety Oakden-Rayner et al. (2020). Therefore, it is critical to design training algorithms that can enhance robustness of segmentation methods against input perturbations.

A significant number of researches focusing on the robustness of computer vision have been conducted in recent years Arnab et al. (2018) Kamann & Rother (2020) Robey et al. (2020) Tramer et al. (2020). However, the majority of existing works focus on image classification tasks Hendrycks & Dietterich (2019). Robust semantic segmentation methods against adversarial attacks have been proposed in Hsin et al. (2019) Xue et al. (2018) Hung et al. (2018) Xie et al. (2017). Work conducted by Goodfellow et al. (2015) proved that neural networks can be easily misled by some intentionally designed yet imperceptible perturbations to generate an incorrect answer with high confidence. In computer vision, adversarial attacks apply pixel-level changes onto the image that lead the model to wrong predictions, where the attacked image looks perceptually similar to the original one Ma et al. (2020). Perturbation-based robust training algorithms have already tackled this problem. Nevertheless, in real life, there may be changes that raised by some out-of-distribution variations such as snow weather or extreme brightness that can not be represented using small pixel-level changes.

Current works on robustness mainly focus on adapting the domain gap on a single side, i.e., either from image or label side. For instance, Robey et al. (2020) proposed a model-based robust learning architecture which is applied on Convolutional Neural Networks(CNN) to maintain high prediction accuracy under natural variations for image classification tasks. Yet it is highly dependent on the performance of the natural variation model that capture the changes from source to target domain,

which makes the algorithm less robust and less scalable in semantic segmentation task. Meanwhile, AdaptSegNet proposed by Tsai et al. (2020) minimizes the distribution gap on the output side. In this paper, we build upon Robey et al. (2020) to design a new robust training algorithm for semantic segmentation tasks. The objective is to build a segmentation model that generates high accuracy predictions under natural variation effects. Our method also utilize the idea from Tsai et al. (2020) that images from different domains with great appearance difference may share some similarity on the label side such as spatial layout and local context. We showed that the semantic feature map of an image under any natural variations remains unchanged which we refer to as semantic meaning invariance. Our proposed training algorithm minimizes the gap on both image and label side. On image side, the model-based robust training algorithm is applied to train a model using the simulated target domain images to enhance robustness. On label side, we apply generative adversarial networks (GANs) to minimize the feature map gap between simulated natural variation images and target domain images. We have also included extensive comparisons showing that our method outperforms related state-of-the-art works in domain adaptation.

Our contributions are: 1) We propose MRTAdapt, a new model-based training algorithm to enhance robustness of DNN-based semantic segmentation methods against natural variations. 2) We build on top of generative adversarial networks and model-based robust training algorithms to minimize the gap on both image and label side to enhance robustness. 3) Our results on Cityscapes Cordts et al. (2016) and Synthia Ros et al. (2016) datasets show that our method outperforms multiple state-of-the-art domain adaptation techniques, such as AdaptSegNet Tsai et al. (2020), ADVENT Vu et al. (2019) and FDA Yang & Soatto (2020).

## 2 RELATED WORK

**Semantic Segmentation.** In the past decades, Convolution Neural Networks(CNN) are widely used in semantic segmentation. Current state-of-the-art semantic segmentation frameworks are mostly developed from Fully Convolutional Network (FCN) by Long et al. (2015). ResNet proposed by He et al. (2015) used a residual block to sum the nonlinear activation output and identity mapping, which is proved to improve the gradient propagation and increase the accuracy of semantic segmentation. DenseNet Huang et al. (2018a) builds upon ResNet and uses the concatenation of previous feature maps called dense block. This gives each layer in DenseNet information from all preceding layers. Jégou et al. (2017) extended DenseNet into FCN architecture. SegNet proposed by Badrinarayanan et al. (2017) introduces the deep convolution encoder-decoder architecture to the field of semantic segmentation. Also, in Zhou et al. (2015), it is proved that empirical size of receptive field is much smaller than the theoretical size. Chen et al. (2016) and Yu & Koltun (2016) used dilated convolution to enlarge the receptive field. ParseNet by Liu et al. (2015) adds global context to CNNs for semantic segmentation. He et al. (2014) introduced spatial pyramid pooling in DCNN. PSPNet introduced by Zhao et al. (2017) uses a novel global pyramid pooling module to capture both global context information. Duta et al. (2020) extended the idea by combining both local and global Pyramidal Convolution blocks in the neural network model. Chen et al. (2017b) introduced atrous spatial pyramid pooling. Depth image is also used along with original RGB information for semantic segmentation by Wang et al. (2019). For actual applications, Zhao et al. (2018) proposed ICNet for real-time semantic segmentation accomplishing fast inference without sacrificing too much quality left behind. Azimi et al. (2020) proposed aerial perspective dataset for dense semantic segmentation.

**Domain Adaptation.** Combining with the techniques of Generative Adversarial Network (GAN) Goodfellow et al. (2014), Ganin & Lempitsky (2015) proposed DANN to reduce the distribution gap between different domains by using discriminator to make the prediction cannot be identified between source and target domain. Pan et al. (2020) first separate target domain into splits based on entropy-based ranking and then deploy self-supervised adaptation technique to reduce the domain gap between synthetic data and real images. PIT proposed by Lv et al. (2020) constructs pivot information shared across domains. Chen et al. (2019) used depth image as guided information to build adaptation method from synthetic to real dataset. In the field of utilizing synthetic datasets to auxiliate the training process of real-world images, MUNIT proposed by Huang et al. (2018b) learn conditional distribution of target domain which can separate domain-invariant semantic content of an image from domain-specific properties. Vu proposed ADVENT model which maximize prediction certainty in target domain by introducing entropy loss Vu et al. (2019). Zhu proposed CycleGAN Zhu et al. (2020) which was trying to learn a mapping such that the distribution of generated image is

indistinguishable from the target domain distribution using adversarial loss. AdaptSegNet proposed by Tsai et al. (2020) aimed to reduce the gap between the outputs from source and target domain given that images might be very different in appearance. FDA proposed by Yang & Soatto (2020) uses Fast Fourier Transform(FFT) to adapt source and target domain.

**Robustness.** As for robustness against corrupted images, Hendrycks & Dietterich (2019) established rigorous benchmarks for image classification and proposed a series of image corruption examples. Kamann & Rother (2020) showed that robustness increases with the performance of the semantic segmentation model and dense prediction cell was only designed to improve performance on clean data. Regarding robustness on semantic segmentation, Arnab et al. (2018) did the first evaluation of adversarial attacks on semantic segmentation and analyzed multi-scale processing. Robey et al. (2020) proposed model-based robust training architecture focusing on the topic of image classification. Wong & Kolter (2020) bridged the gap between real-world perturbations and adversarial defenses by learning perturbation sets from data, through common image corruptions.

## 3 METHOD

In this section we propose a new robust training algorithm for semantic segmentation tasks. Particularly, in section 3.1, we define the semantic segmentation task and the natural variations. In section 3.2, we give the outline of our proposed algorithm. In section 3.3, we define semantic meaning invariance and clarify its connection to our algorithm. In section 3.4, we introduce the detailed training procedure of our model-based robust adaptive training algorithm(MRTAdapt).

### 3.1 SEMANTIC SEGMENTATION AND NATURAL VARIATION

Consider an input $x \in \mathbb{R}^{H \times W \times 3}$ representing an RGB image with three channels and its corresponding label map $y \in \mathbb{R}^{H \times W}$ which is annotated for every pixel in the image. A semantic segmentation task is to train a neural network that generates pixel-wise segmentation feature map $\hat{y} \in \mathbb{R}^{H \times W}$. We assume the dataset is drawn $i.i.d.$ from distribution $(x, y) \sim \mathbb{P}$. The optimization objective in semantic segmentation is to find the best weight $w$ that minimizes the loss function $\mathcal{L}(x, y; w)$ with respect to input $x$, label $y$ and weight $w$, which can be written as:

$$\min_w \frac{1}{n} \sum_{j=1}^{n} \left[ \mathcal{L}(x, y; w) \right] \tag{1}$$

Next we denote by $\delta \in \Delta$ a natural variation applied to $x$ that is derived from a nuisance space $\Delta$. A model of natural variation can be learned from the difference of source and target domain images. Particularly, we denote the natural variation model that transforms the input image $x$ from the source domain into an image in the (perturbed) target domain as $V(x, \delta)$. Note that, in general, it is hard to write a natural variation model learned from data into an analytical form because there is no closed form expression capturing the natural perturbation. Natural variation model can be customized regarding the needs of each dataset. In this paper, our goal is to design a semantic segmentation method that is robust to natural variations $\delta$.

### 3.2 ALGORITHM OVERVIEW

Our proposed model-based robust adaptive training algorithm (MRTAdapt) shown in Figure 1 has two modules: a segmentation network **G** using a model-based robust training algorithm to enhance robustness on image side, and a discriminator network **D** to enhance robustness on label side. Two sets of images from source and target domain will have the notation of $I_s$ and $I_t$, while the image generated using natural variation model will be denoted as $I_{nv}$. After using source domain images to train the model, we transform images from source domain with natural variation model. Then we use simulated natural variation images to find the highest segmentation loss by applying the model-based robust training algorithm to optimize the segmentation network. Then we predict the segmentation softmax output for the unlabeled target domain images $I_t$ by forwarding the target images into the same segmentation network. To minimize the gap between the output of natural variation images and target domain images, we use these two predictions as the input to the discriminator to distinguish whether the feature map generated by segmentation network is from natural variation model or target

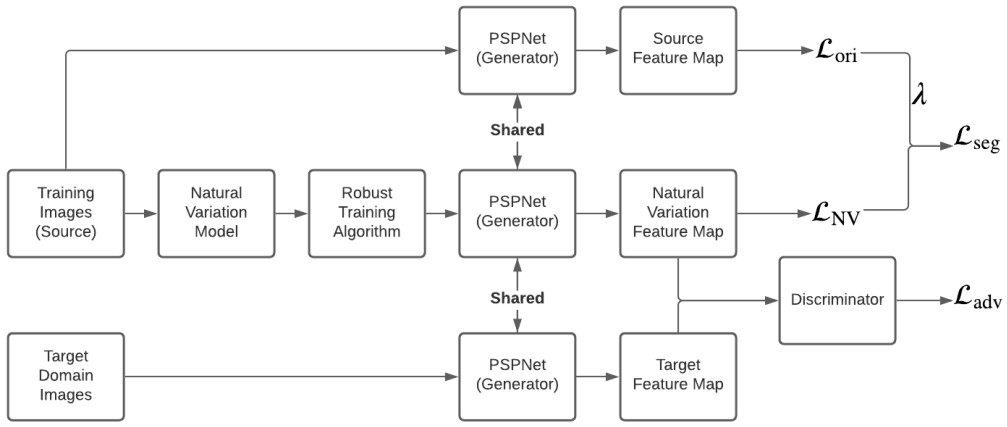

Figure 1: Model-Based Robust Adaptive Training Algorithm (MRTAdapt)

domain. Using an adversarial loss for discriminator, the loss will back-propagate to segmentation network to help segmentation network to generate similar distributions in target domain to natural variation. Detailed information will be covered in the following sections.

### 3.3 SEMANTIC MEANING INVARIANCE

Images under different kinds of natural variations have huge visual perspective difference. For domain adaptation problem in computer vision, the motivation of most works is based on the fact that annotating images in a new domain setting is time-consuming and extremely expensive. In this paper, we are trying to simulate the real-world setting when the model is trained with annotated images under normal condition in the source domain but natural variation conditions from target domain occur during inference time. Most of the current works focus on improving the performance from the image side. AdaptSegNet tries to solve this problem from the prospective of the label side. The experiment conducted in this work has proved that spatial layout and local context of labels are similar for different datasets for semantic segmentation problem Tsai et al. (2020).

Inspired by the idea of minimizing the gap on label size, we found the semantic information will remain the same from the label perspective regardless of the types of natural variation on the image. For example, the car in the image under strong brightness will remain the semantic meaning as a car compared to normal setting. Take Figure 2 as an example, left side is the same image with different brightness condition compared to the right side. Since the image is only changed into a brighter version while all the objects remain the same, the semantic information will be unchanged under natural variation. In the context of semantic segmentation, unchanged semantic information means labels will be the same. Thus, labels in the feature map will remain unchanged when we apply natural variation model on source domain image to transform into target domain.

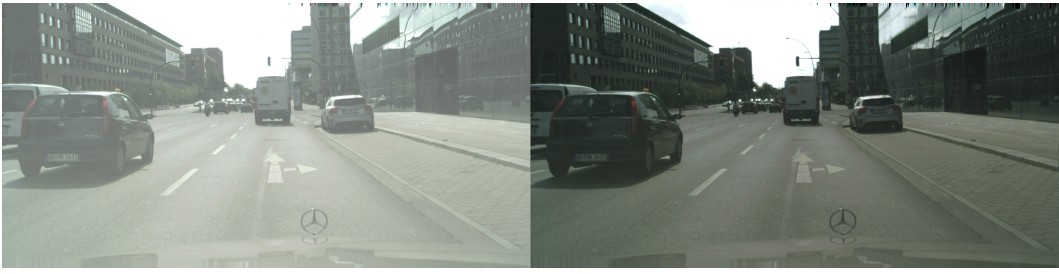

Figure 2: Semantic meaning of an object will not change with natural variation

### 3.4 MODEL-BASED ROBUST ADAPTIVE TRAINING ALGORITHM

Different from AdaptSegNet where the goal was to minimize the gap between the label from source domain and target domain, we build on top of model-based robust training algorithm with different optimization goal. Using the idea of semantic meaning invariance, we treat the generated natural variation image generated as "source" domain image. For the reason that the generated natural variation image is chosen by model-based robust training algorithm is aiming to represent the worst-case scenario of natural variation. The objective of our proposed model-based robust adaptive training algorithm is trying to reduce the gap on both image and label side.

Here we formulate the optimization task as a min-max problem, which will be minimizing the segmentation loss for source and natural variation images, while maximizing the probability of natural variation predictions being treated as target domain feature map. Our proposed algorithm is an integration of model-based robust training algorithm and generative adversarial network.

$$\max_{\mathbf{D}} \min_{\mathbf{G}} \mathcal{L}(I_s, I_t, I_{nv}) \tag{2}$$

#### 3.4.1 MODEL-BASED ROBUST TRAINING FORMULATION

The objective of the model-based robust semantic segmentation algorithm is to train the model so that it can be robust to natural variations such changes in the weather or lightning conditions. Given that we only have labeled data for source domain images while labels for target domain are unavailable, our goal is to simulate and transform the normal condition training data into natural variation cases and then, the model can be trained until it generates high accuracy prediction under abnormal conditions. In general, the concept of this algorithm can be treated as a data augmentation method that widens the range of conditions under which the model can attain high performance. The objective of the robust model-based training algorithm can be formulated as Robey et al. (2020):

$$\mathcal{L}_{\text{nv}} = \min_{w} \frac{1}{n} \sum_{j=1}^{n} \left[ \max_{\delta \in \Delta} \mathcal{L}\big(V(x, \delta), y; w\big) \right] \tag{3}$$

The inner maximization represents the loss inferred by the natural variation model. Given an input image x from source domain, we need to search in the nuisance space $\Delta$ for a local optimal parameter $\delta$ that generates the worst-case scenario of natural variation. The optimal nuisance parameter is denoted as $\delta^*$ that infers biggest loss between prediction and original label map. The objective of the inner maximization problem is to maximize the segmentation loss of the model under natural variation. The objective of outer minimization problem is to train the weights of the model $w$ to be robust against worst-case natural variation condition upon convergence, which is represented in the form of minimizing the sum of the loss in this case.

During actual implementation, we assume the number of datapoints being trained is a finite number $N$. This means a dataset being recognized as training set will contain N image-label pairs: $\{x_i, y_i\}_{i=1}^{N}$. Also, the natural variation model being applied in the algorithm is learned from data. Model-based Robust Training generates random domain-specific style code for input images from source domain and filters the style codes which infer maximal loss out for weights update. That means, for every mini-batch size of input images $(x, y)$ from source domain, we apply the natural variation model to transform the image into target domain, where we will treat the transformed image and original label as a new image-label pair denoted as $(V(x, \delta), y)$. Then we utilize both pairs of datapoints to train the network for the outer minimization problem.

#### 3.4.2 NETWORK TRAINING PROCEDURES

For the segmentation network $\mathbf{G}$, we define the segmentation loss as a weighted sum from both the loss inferred from source domain images as well as the highest loss derived from natural variation images using model-based robust training algorithm. Two losses are connected via $\lambda_{\text{nv}}$ and $\lambda_{\text{ori}}$, which represent the contributions of generated natural variation images and source domain images to the segmentation network respectively:

$$\mathcal{L}_{\text{seg}}\ (I_s, I_{nv}) = \lambda_{\text{nv}}\mathcal{L}_{\text{nv}} + \lambda_{\text{ori}}\mathcal{L}_{\text{ori}} = \min_{w} \left[ \lambda_{\text{nv}} \max_{\delta \in \Delta} \mathcal{L}(V(x, \delta), y; w) + \lambda_{\text{ori}}\mathcal{L}(x, y; w) \right] \tag{4}$$

---

**Algorithm 1** Model-Based Robust Training Algorithm

---

1: $\delta^* \leftarrow 0, L_{\max} \leftarrow 0$
2: **foreach** Batch $(x, y)$ **do**
3:    **for** k steps **do**
4:       Sample $\delta$ from $\Delta$ uniformly random
5:       $L_{\text{current}} \leftarrow \mathcal{L}(V(x, \delta), y; w)$
6:       **if** $L_{\text{current}} > L_{\max}$ **then**
7:          $L_{\max} \leftarrow L_{\text{current}}$
8:          $\delta^* \leftarrow \delta$
9:       **end if**
10:   **end for**
11:   $\nabla \leftarrow \nabla_w(\lambda_{nv}\mathcal{L}(V(x, \delta^*), y; w) + \lambda_{ori}\mathcal{L}(x, y; w))$
12: **end**

---

For target domain images, we feed them into segmentation network and obtain an adversarial loss, which aims to fool the discriminator and maximize the probability of target domain predictions $P_t$ being considered natural variation predictions. For the reason that we do not have the ground truth feature map of target domain images, segmentation loss will not be calculated.

$$\mathcal{L}_{adv}(I_t) = -\sum_{h,w} \log\left(\mathbf{D}(P_t)^{(h,w,1)}\right) \tag{5}$$

For the discriminator training, we forward a given segmentation softmax output $P$ to a fully-convolutional discriminator using cross-entropy loss for two classes (natural variation and target domain). The loss can be written as:

$$\mathcal{L}_d(P) = -\sum_{h,w}(1-z)\log\left(\mathbf{D}(P)^{(h,w,0)}\right) + z\log\left(\mathbf{D}(P)^{(h,w,1)}\right) \tag{6}$$

So the min-max problem can be rewritten into the following form, where segmentation loss $\mathcal{L}_{seg}$ and adversarial loss $\mathcal{L}_{adv}$ is summed by applying corresponding weights representing the contribution to the network:

$$\max_{\mathbf{D}} \min_{\mathbf{G}} \left[\lambda_{seg}\mathcal{L}_{seg}(I_s, I_{nv}) + \lambda_{adv}\mathcal{L}_{adv}(I_t)\right] \tag{7}$$

## 4 IMPLEMENTATION

### 4.1 DATASET PREPROCESSING

**Brightness** Characteristic of an image can be represented using Hue-Saturation-Value (HSV), where hue, saturation and value represent the color, grayness and brightness respectively Latifah et al. (2020). In this paper, we use natural variation model to simulate target domain images. To achieve a better simulation result, we arrange our data preprocessing for brightness condition into two phase: Phase 1: divide original dataset into three subsets: bright, medium and dark by calculating the average Hue-Saturation-Value of each image. In this paper, we divide the three subsets by their HSV values with 72 and 90 representing the boundary of dark-medium and medium-bright on Cityscapes dataset, using 75 and 130 for Synthia dataset. For training set, use bright subset as source domain and dark subset as target domain to simulate the fact that most current dataset is constrained in day-time condition. Dark subset is referring to the case when a vehicle is moving at nighttime under low brightness condition. There are 931 adjusted source domain images representing bright condition, and 486 target domain images representing dark condition for Cityscapes dataset. For Synthia dataset, bright and dark images are 1200 and 1174 respectively. Phase 2: adjust the overall brightness of bright subset by applying OpenCV library to enlarge the difference of brightness between training set and testing set.

**Snow** For snow weather as natural variation, we eliminate the effect of brightness and rearrange the training set with different configuration. For Cityscapes, we have source domain subset with 1046 images and target domain subset with 1064 images. For Synthia, source domain and target domain have 1000 and 1170 images respectively. Both subsets contain images in bright and dark condition, which allows the network to be exposed to images from different brightness condition. So the natural variation of brightness effect discussed in previous section is minimized. We remained unchanged for source domain images, referring to the case that we train our model in normal condition. Then we apply the snow corruption operation being used in ImageNet-C with severity of 1 to target domain images, referring to the case that our model would be exposed to natural variation images during inference. Due to the limitation of computing resources, the image size being trained in this paper is relatively small. So we apply a proper level of severity for snow effect so that the performance will not break down completely.

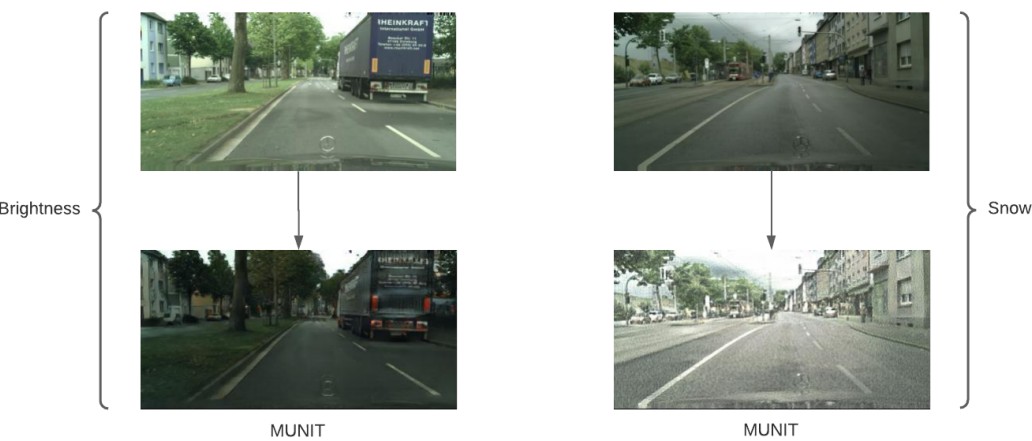

Figure 3: Example frames of both natural variations

## 4.2 IMPLEMENTATION DETAILS

In this paper, natural variation model is powered by MUNIT Huang et al. (2018b) on Cityscapes dataset and RAIN proposed by Luo et al. (2020) on Synthia dataset.

**PSPNet** With the natural variation model derived from either known a priori or model learning from data. For the latter one, we apply MUNIT on Cityscapes dataset and RAIN on Synthia dataset, we use PSPNet Zhao et al. (2017) to do semantic segmentation. In PSPNet, we use ResNet He et al. (2015) model as backbone with the dilated network method Yu & Koltun (2016), then plug in the robust training module to generate new datapoints and obtain feature map using the weighted combination. Pyramid pooling module with 4 level representing different size of the sub-regions in the feature map is applied to gather both local and global region information. Then we concatenate the feature map obtained by pyramid pooling module with original feature map to generate final feature map.

**MUNIT** For MUNIT Huang et al. (2018b) model that captures natural variation model, we set the training iterations into 300000. Weight decay is set to 0.0001 with image size fixed at $200 * 200$. The architecture of MUNIT is consisted of two auto-encoders, one for each domain. For each auto-encoder, it is composed by one content encoder, one style encoder and one decoder. Content encoder is consisted of strided convolutional layers for down-sampling purposes followed by residual blocks. Instance normalization is attached at the end of each convolutional layer in content encoder. Style encoder first down-samples the images by convolutional layer, then global pooling and fully connected layer are applied. For the decoder, style code being generated is fed to a multilayer perceptron to generate parameters for Adaptive Instance Normalization (AdaIN). The AdaIN is applied to the residual blocks which are processing the content code. The output will then be up-sampled and serve as the final reconstructed image. LSGAN is serving as discriminator while VGG is applied for the domain-invariant perceptual loss calculation.

**RAIN** For RAIN Luo et al. (2020) model proposed by Luo. We set the training iterations to 100K and learning rate as 0.0001. This model is the composition of AdaIN Huang & Belongie (2017) and auto-encoder. For AdaIN, VGG model is applied as encoder.

## 5 EXPERIMENT

Due to computation resources limitation, the experiments in this paper are done on Google Cloud Platform (GCP) using 16 GB of memory and one NVIDIA Tesla T4 GPU with 16 GB GDDR6 of graphic memory. The operating system being used in the experiment is Ubuntu 18.04. The code is implemented based on Python 3.6 and PyTorch 1.5.1. For fair comparison, all the images being trained in this paper will have the same size of $201 * 201$ with batch size of 8 for all the methods being applied in the experiment. For evaluation metrics, we use pixel-wise accuracy and mean of class-wise intersection over union (mIoU) to define evaluation metrics. Pixel-wise accuracy is calculated using the number of pixels common between prediction output and ground truth divided by the number of pixels in ground truth feature map. The Intersection over Union metric is also named as Jaccard index to calculate the overlap between prediction and ground truth. To be specific, IoU (IoU $= \frac{\text{Area of Overlap}}{\text{Area of Union}}$) calculates the number of pixels common between prediction output and ground truth divided by number of pixels across both feature maps. Inspired by Zhao et al. (2017), we use ResNet-101 to generate basic feature map. ResNet-101 keeps a proper balance between high performance and proper complexity. Serving as baseline of the experiment, we train and test PSPNet using source domain images as training set and test the performance in target domain. Source domain only PSPNet tends to present how natural variation affects the performance in normal configuration. To compare our model in a comprehensive way, we also evaluate the performance of PSPNet when it is trained using labeled images from target domain (dark condition) and test the performance using images from target domain. We train all models for 200 epochs with basic learning rate of 0.01. For the MUNIT network being trained in this experiment, we use batch size of 1, weight decay of 0.0001 and learning rate of 0.0001 with step side of 10000 referring to the frequency of learning rate decaying.

**Comparison Methods** We compare our method with several recent domain adaptation methods, including AdaptSegNet Tsai et al. (2020), ADVENT Vu et al. (2019), and FDA Yang & Soatto (2020). AdaptSegNet introduces generative adversarial network technique to reduce the gap on the label side of source domain and target domain. Two methods are proposed in ADVENT, namely MinEnt and AdvEnt. MinEnt maximize prediction certainty in the target domain by using a proposed entropy loss $\mathcal{L}_{ent}$. AdvEnt shares similarity with the concept of AdaptSegNet and minimize the entropy by making target entropy distribution similar to the source domain. We evaluate the performance of both methods being proposed by ADVENT in this paper. FDA utilize Fast Fourier Transform (FFT) and replace the low-level frequencies of target domain images into source domain images. From the original literature of these proposed methods, the semantic segmentation baseline being applied is DeepLabV2 Chen et al. (2017a) with ResNet101 using SGD. To compare the performance from the same backbone of segmentation network, we replace the DeepLabV2 into PSPNet in this paper. All the configurations including initial learning rate of 0.1, momentum of 0.9 and weight decay of 0.0001 keep unchanged.

**Cityscapes**

From Table 1, segmentation model that only trained with source domain images will have huge performance drop during the inference time under natural variations. Model-based robust training algorithm (MRT) has 11% and 7% of improvement in terms of IoU compared to the model only trained with source domain images. The other domain adaptation methods such as AdaptSegNet, AdvEnt and MinEnt have stable and better performance compared to the model-based robust training algorithm. FDA method has 15% of improvement in mIoU compared to source only PSPNet for brightness case yet not promising result on snow effects. Our proposed method has 16% and 12% of mIoU increase compared to baseline, which outperforms all other comparison methods. For brightness condition, our proposed MRTAdapt even has higher accuracy than segmentation model trained with target domain images.

**Synthia** For Synthia dataset, we apply RAIN model in stead of MUNIT model to capture natural variation effect. As results shown in Table 2, comparing with all other methods, our proposed MRTAdapt outperforms other state-of-the-art domain adaptation models and also reach similar per-

| | Brightness | | | Snow | | |
|---|---|---|---|---|---|---|
| method | mIoU | mAcc | AllAcc | mIoU | mAcc | AllAcc |
| PSPNet(Source) | 0.2221 | 0.2663 | 0.7801 | 0.2230 | 0.2731 | 0.7785 |
| MRT | 0.3384 | 0.4037 | 0.8642 | 0.2928 | 0.3511 | 0.8372 |
| AdaptSegNet | 0.3509 | 0.4061 | 0.8476 | 0.3222 | 0.3849 | 0.8356 |
| AdvEnt | 0.3570 | 0.4153 | 0.8714 | 0.3091 | 0.3787 | 0.8250 |
| MinEnt | 0.3536 | 0.4088 | 0.8642 | 0.3254 | 0.3947 | 0.8492 |
| FDA | 0.3706 | 0.4308 | 0.8804 | 0.1892 | 0.2348 | 0.7175 |
| MRTAdapt(Ours) | 0.3804 | 0.4453 | 0.8805 | 0.3443 | 0.4108 | 0.8591 |
| PSPNet(Target) | 0.3753 | 0.4288 | 0.8922 | 0.3936 | 0.4466 | 0.8959 |

Table 1: Performance Comparison for Brightness and Snow on Cityscapes

formance to the case when we have the ground truth label of target domain. Meanwhile, our method shows high robustness in different abnormal conditions.

| | Brightness | | | Snow | | |
|---|---|---|---|---|---|---|
| method | mIoU | mAcc | AllAcc | mIoU | mAcc | AllAcc |
| PSPNet(Source) | 0.4910 | 0.5741 | 0.8567 | 0.2675 | 0.3436 | 0.7490 |
| AdaptSegNet | 0.4506 | 0.5278 | 0.8555 | 0.2471 | 0.3279 | 0.7367 |
| AdvEnt | 0.3250 | 0.4018 | 0.7956 | 0.3503 | 0.4322 | 0.8272 |
| MinEnt | 0.3049 | 0.3944 | 0.7698 | 0.2526 | 0.3280 | 0.6990 |
| FDA | 0.5888 | 0.6521 | 0.9159 | 0.2895 | 0.3758 | 0.7621 |
| MRTAdapt(Ours) | 0.6861 | 0.7577 | 0.9430 | 0.4516 | 0.5292 | 0.8788 |
| PSPNet(Target) | 0.7178 | 0.7076 | 0.9540 | 0.3173 | 0.4144 | 0.7494 |

Table 2: Performance Comparison for Brightness and Snow on Synthia

## 6 CONCLUSION

In this paper, we addressed the problem of fragility in current learning-based semantic segmentation methods against natural variation effects that typically occur in real-world conditions. Using industry-level segmentation architecture PSPNet as backbone network, we adopted a model-based robust training algorithm that was originally proposed for image classification problem to design a robust semantic segmentation method. By applying a model which can capture the changes that lie within source and target domain, we challenge the network with the worst-case scenario natural variation images by finding the biggest loss inferred. Also, building upon the AdaptSegNet that minimizes the distribution gap of source and target domain from the label perspective, we showed that natural variation will not alter the semantic meaning of the label. Based on this idea, we proposed a model-based robust adaptive training algorithm that achieves higher performance than several domain adaptation methods do. Our future research directions will focus on reducing and eventually eliminating the dependence of the proposed method on the performance of the model that captures natural variation effects

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

# 7 APPENDIX

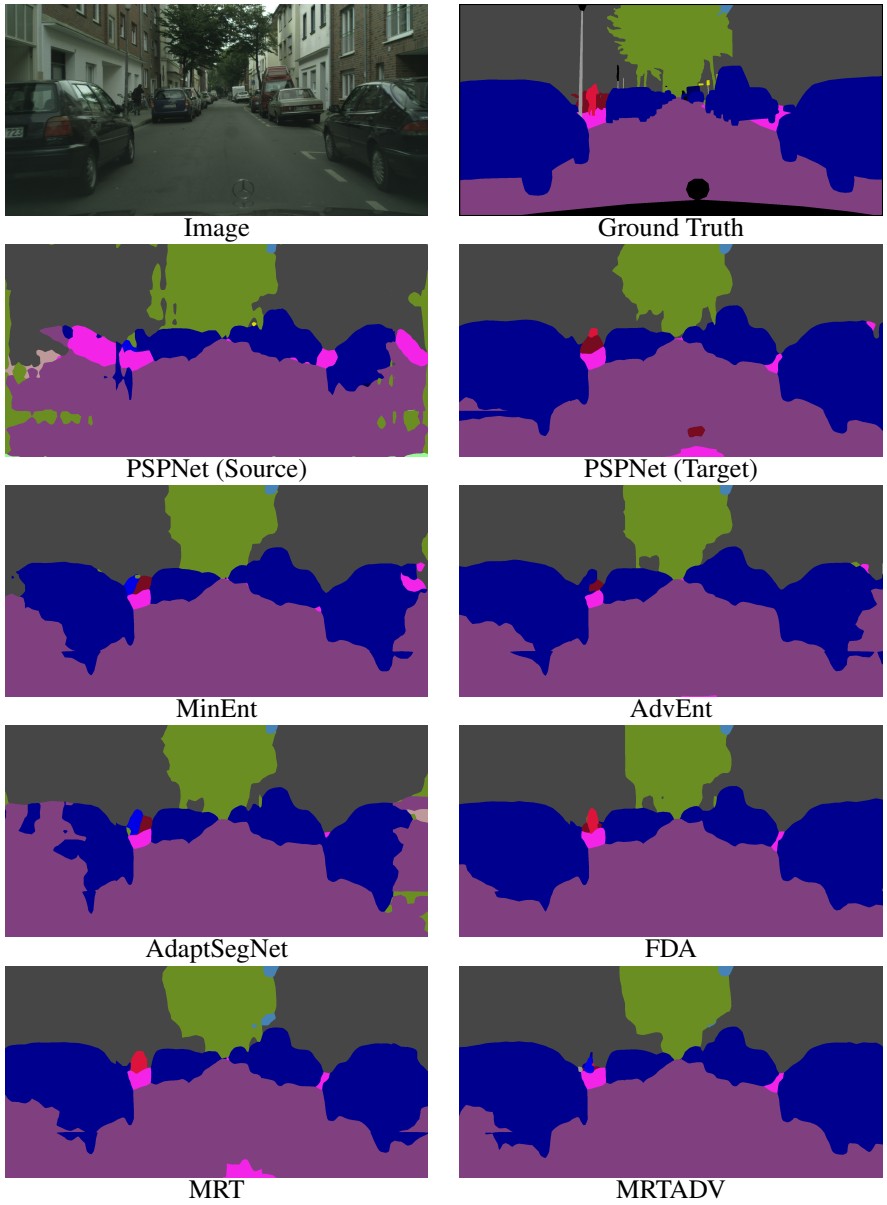

Table 3: Brightness Test Example - Cityscapes

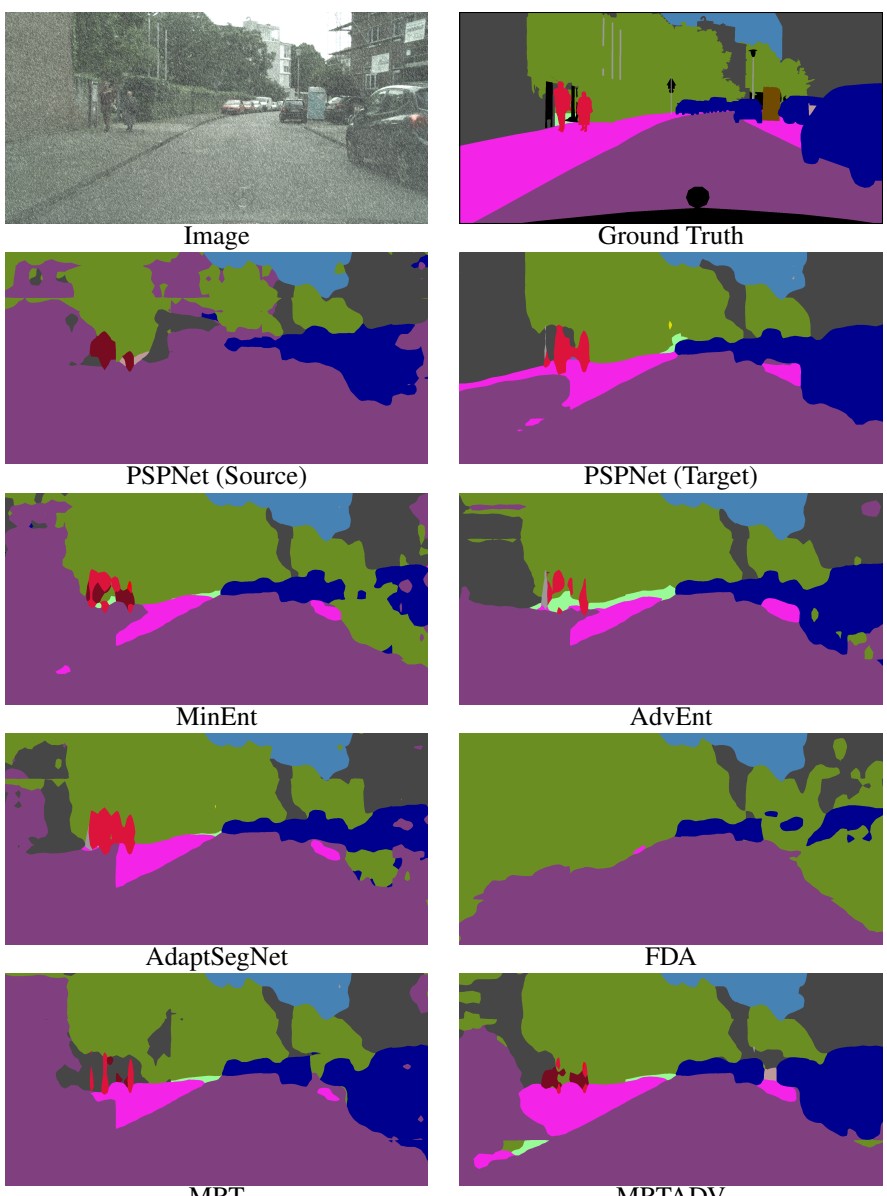

Table 4: Snow Test Example - Cityscapes

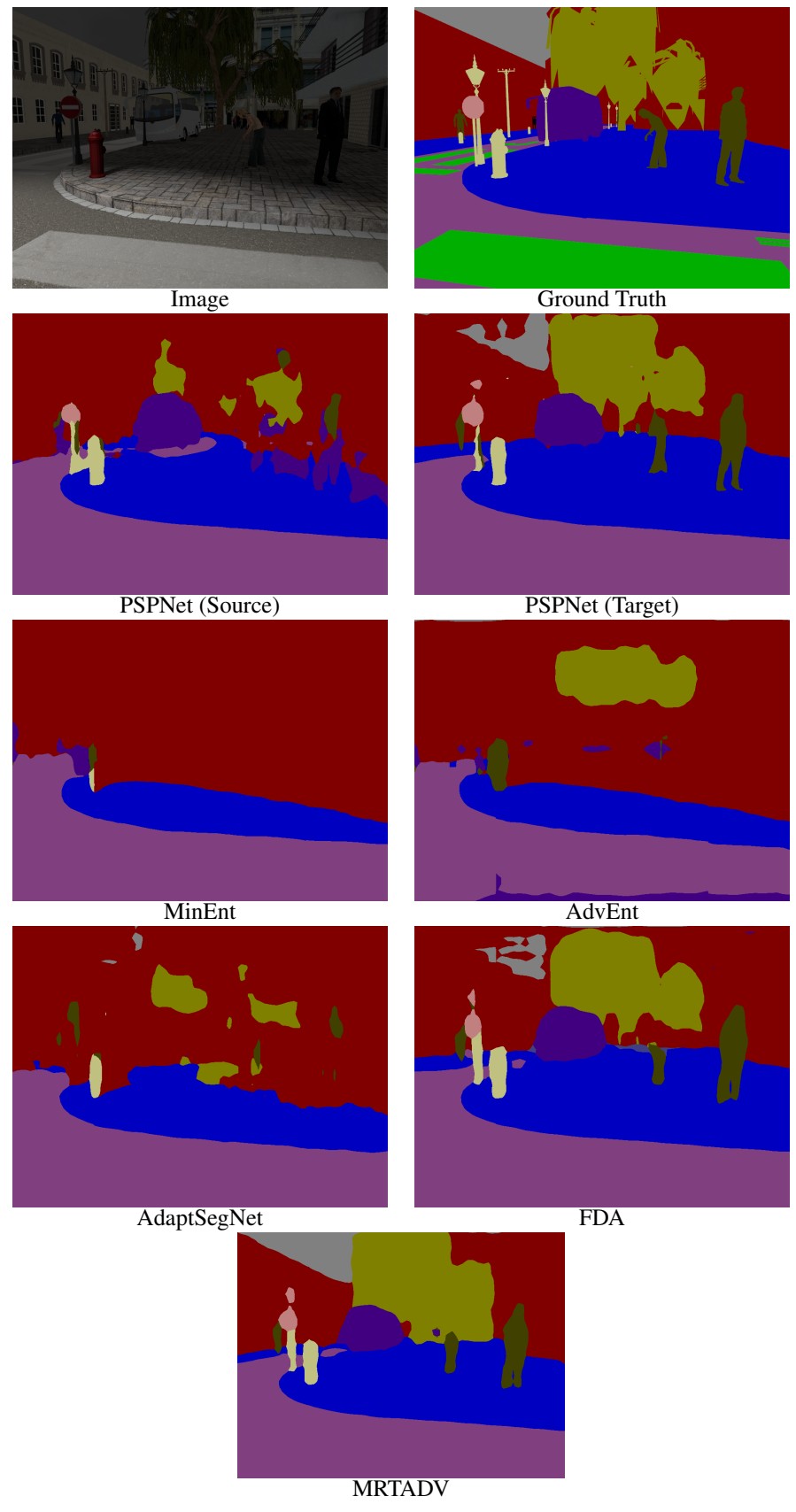

Table 5: Brightness Test Example - Synthia

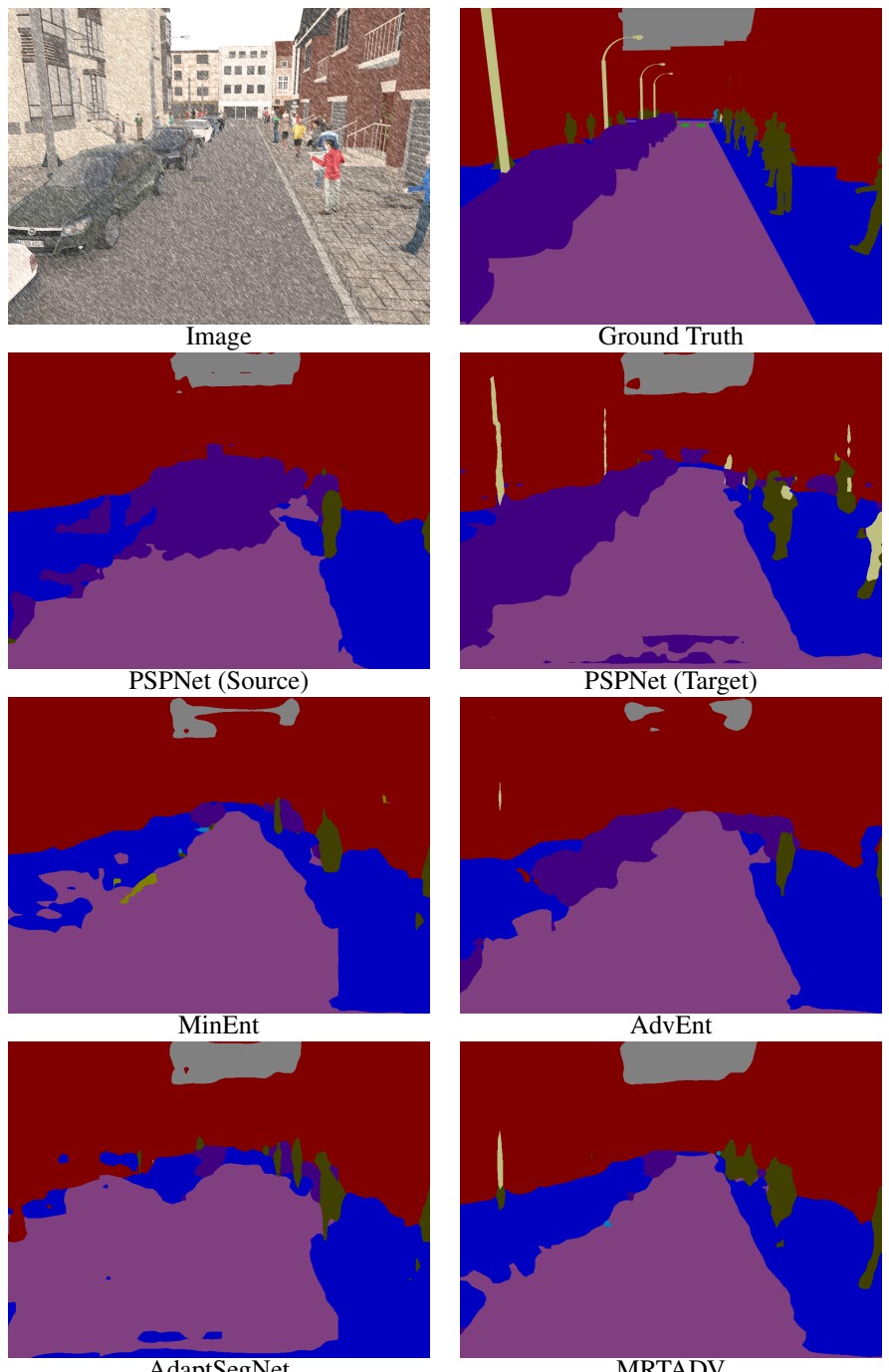

Table 6: Snow Test Example - Synthia

