# OpenReview forum: "Model-Based Robust Adaptive Semantic Segmentation"
_ICLR.cc/2022/Conference — ICLR 2022 Submitted_

### Official Review · Reviewer_9pKw · 2021-11-01

**Correctness:** 2
**Technical Novelty And Significance:** 2
**Empirical Novelty And Significance:** 2
**Recommendation:** 5
**Confidence:** 4

**Main Review:**

--Strengths--

The problems being addressed here are real, and are important -- principled solutions and improvements for model robustness with respect to semantic image segmentation are of high value to the community and additionally likely result in improved performance in many practical applications. I encourage the authors to think about these problems. Global structure of the manuscript is reasonable and system overview figures and pseudocode are appreciated, towards aiding understanding.

--Weaknesses--

The paper considers an interesting problem (segmentation model robustness) yet unfortunately the composition would appear in a premature state making it difficult to fully grasp the size of the novel contributions. Phrasing issues and sloppy writing currently distract the reader.

There has been previous work on analyzing the influence of synthetic data quality and usage strategies for semantic segmentation; eg. Zhang et al. (2017).  Additional works also investigate which properties of synthetic training data help generalization to real data, across various tasks. See eg. Su et al. (2015), Movshovitz-Attias et al. (2016). Experimental evidence demonstrating the efficacy of how well synthetic data strategies can generalize to real-world data and imagery are often key to the strength of such works. I believe analogous investigation here would result in more convincing evidence towards the value of the approach. The current lack of such experimental validation leaves the submission somewhat lacking.

Minor (non-exhaustive):

* Section 2 of the manuscript makes an attempt at surveying related work however handfuls of previous papers are each dedicated a terse summary sentence, which may leave the reader somewhat unengaged. Explicitly contrasting these previous works with novel insight towards the problem at hand can help to build a compelling story. What are the key component(s) that are missing from the collective previous work? where are their failings? Explaining more clearly the valuable gap that your contributions, will in contrast, fill can serve to both strengthen the work and help your audience.

* Contradictory statements serve to confuse the reader eg.

 'lack of robustness prevents the application of learning-based semantic segmentation methods on safety-critical applications' ; and

'There are many applications now being used have made great progress with the help of semantic segmentation, such as medical image processing [..], autonomous vehicles [..] and robotics [..].' (pg. 1).

* Table row usage of bold font appears unconventional and rather autocratic.

**Summary Of The Paper:**

This paper considers the problem of semantic image segmentation and endeavours to improve the performance of network-based approaches by enhancing model robustness to specific causes of image-space perceptual variance (eg. weather, illumination brightness). A training strategy that leverages generative (GAN) based components, data augmentation and domain adaptation ideas is employed towards encouraging models to represent generated images and target domain images, similarly. Resulting quantitative performance is reported across two standard datsets where comparisons are drawn with recent alternative works.

**Summary Of The Review:**

As stated, the direction is valuable and the ideas somewhat interesting however I feel this submission is currently in a premature state. Lack of clear exposition and thorough real-world experimental evidence weigh rather heavily in the rating. I encourage authors to work on some of the methodological, presentation suggestions.

---

> ### Author Response · Authors · 2021-11-18
> **Related works**
>
> We thank the review for their suggestions. We agree with this review that analogous investigation regarding the synthetic data for semantic segmentation would result in more convincing evidence towards the value of the approach. We would like to note that the domain adaptation methods being applied to do comparison in this paper, including AdaptSegNet, AdvEnt, FDA were all designed to use labeled synthetic datasets to improve the accuracy of the model on real-world settings. Our proposed method borrows partially the strengths of the AdaptSegNet that images have huge different appearances might share similarities on the label side and proposed the concept of semantic meaning invariance. This concept shows that images under any kind of natural variations will share same semantic feature labels. The proposed concept build the robustness of the model on the label side. Along with the concept from the classification-based robust training method proposed by Robey, our robust training method is aiming to solve the robustness of data on natural variations on both image and label side.
> We would like to highlight that in the modified manuscripts, we have added some more literature reviews in the field of the evaluation on synthetic data to real-world data in the field of semantic segmentation. In the near future, we will keep working on more datasets with enough actual natural variations for training.

---

> ### Author Response · Authors · 2021-11-18
> **Section 2 of the manuscript**
>
> We thank the reviewer for their detailed comments and suggestions. We would like to highlight that in our modified manuscripts, we have modified section 2 (related work) to emphasize on the works have strong connection to the problems we are trying to solve in this paper.
> Furthermore, to elaborate the main contribution of our work, the existing works are focusing on minimizing the gap from a single perspective, either image side or label side. Works proposed by [Tsai et al 2020], [Vu et al 2019] are trying to apply Generative Adversarial Network (GAN) to narrow the gap of the segmentation outputs. Works proposed by [Kamann & Rother 2020] and [Robey et al 2020] are focusing on minimizing the gap on the image side by challenging the model with some corrupted data. However, our proposed method combines the strengths from both concepts and create a training method that minimize the gap on both image side and label side and achieved better performance as shown in the experimental results.
> Additionally, we would like to highlight that the closest works to ours are [Robey et al. 2020] and [Tsai et al. 2020]. In the former, a model-based robust training algorithm proposed targeting to solve image-classification problem, which has low-dimension prediction target (usually a number or a name) compared to semantic segmentation problem which has high-dimension prediction feature map that aligns labels for every pixels. Besides, work by Robey is highly dependent on the performance of the natural variation model, which might not be sufficient for semantic segmentation. Meanwhile, in [Tsai et al. 2020] the AdaptSegNet method is proposed based on the observation that images, that are seemingly different, may share similarities such as spatial layout and local context on feature map aspect. A discriminator network is proposed along with the original semantic segmentation network to minimize the distribution gap on label side aiming to help the model generate feature map outputs that look like the ones in target domain.
> Building on top of these two works, our proposed robust training algorithm, called MRTAdapt, aims to minimize the gap on both image and label side. To highlight the difference between our work and [Robey et al. 2020], recall that the latter is applied to image classification problems where the prediction targets are usually low-dimension and small in space. Meanwhile, the model-based robust algorithm proposed in [Robey et al. 2020] is highly dependent on the performance of the natural variation that serves to capture the change in natural variation conditions. These two issues make the algorithm less robust and less scalable in semantic segmentation task, where the prediction target is a high-dimension and complicated feature map. Inspired by the concept presented in [Tsai et al. 2020] that images might share similarities on label side even though their appearance is significantly different on the image side, we proposed a concept named as semantic meaning invariance that the feature map of an image will remain unchanged regardless the change of natural variation conditions. In our proposed method, we apply the robust training algorithm with the help of natural variation model trying to expose the model with the worst performed synthetic target-domain images to enhance the robustness on image size. On the label side, we set up a discriminator to minimize the feature map gap between the output of synthetic target-domain images and true target-domain images to enhance the robustness on the label side. Compared to the method presented in [Tsai et al. 2020] that tries to minimize the gap of the outputs between source and target domain, our method is minimizing the gap between the outputs of synthetic ‘target’ domain and real target domain with the help of the natural variation model. We have also included extensive comparisons showing that our method outperforms related state-of-the-art works including [Tsai et al. 2020] as Table 1 and Table 2 shown in the section 5 (experiment).
> Lastly, we would also want to note that the natural variation model being used in our method can be customized for different datasets for better performance.

---

> ### Author Response · Authors · 2021-11-18
> **Table issues**
>
> We thank the reviewer for their comment. In the revised manuscript, we have revised the tables as per this reviewer’s suggestion.

---

> ### Comment · Reviewer_9pKw · 2021-11-22
> **Post-rebuttal**
>
> I thank the authors for the time spent to provide additional information and comment, which will also be taken into consideration.

---

### Official Review · Reviewer_ChGc · 2021-11-01

**Correctness:** 2
**Technical Novelty And Significance:** 2
**Empirical Novelty And Significance:** 2
**Recommendation:** 5
**Confidence:** 3

**Main Review:**

Strenghts:
+ The work presents a clear motivation for the problem tackled, highlighting the importance of achieving more robust segmentation methods to enable these techniques in real world applications that are very sensitive to mistakes.
+ The introduction makes a clear statement of the contributions of the work
+ Related work presents a nice summary of semantic segmentation models based on deep neural networks.
+ The proposed method obtains better accuracy than other domain adaptation methods in the generated target domain images.

Weaknesses:
- Problem definition. It would be helpful for the reader to have a more explicit statement earlier in the paper (introduction?) about the fact that the authors model the problem of robustness in the segmentation as a domain adaptation problem: from original data to similar data but with “natural variations expected” applied to the original image domain. This is a core idea that can be learned along the paper, but earlier statement would help understand earlier why related work, evaluation, baselines, … are focused on domain adaptation techniques.

- The related work misses details on two relevant aspects.  First, it presents a nice summary of relevant topics (segmentation, domain adaptation, robustness). However, in the domain adaptation, which is key to position the presented work with respect to the existing work, I miss a bit more organised discussion to point what is general related work on the topic and what is focused on the specific semantic segmentation task tackled in this work. Besides, one of the key components in this work is the proposed simulation of natural variations in the scene images, which is a form of image augmentation. Authors mention this relationship with image augmentation briefly in the method description (sec 3), but no mention until then. I think the related work should discuss a bit this topic as well.

- The method needs to be more clearly explained, specially about how the natural variation model and the target domain images are obtained, how do they differ from each other and how realistic the assumptions made about them are.
First, authors mention several times while explaining the method (sec 1 and 3) that “A model for the natural variation can be learned from data”. This is a key assumption (as mentioned in the conclusions), which brings a question: to which extent is this realistic? it assumes that we have enough variation in the target domain data to learn this.  It’s not clearly detailed how this model to apply natural variations is learned. Is it part of the training process when learning the segmentation model? Is the GAN-like training mentioned not only to improve the feature maps generated in the segmentation network but also to learn this image generation? This is not clear in the current description.

Another unclear aspect is about the assumptions on the target domain data. I understood the target domain images are unlabelled. Then, to compare target feature map with natural variation feature map, does the approach assume that those images are aligned somehow with labeled data? i.e., they are different versions (weather, lighting) of the same scene? If this is the case, target domain images are not really unlabelled, since they are aligned to the labeled data. This can be the case in the experimentation, since target domain does not consist of independent images, but images from the training set modified with weather-change simulation effects. It’s not very clear what the assumptions are when reading the method description, one can only see these ideas when reading the experimental set up.

- The experimental validation is not fully convincing on its current form. If I understood correctly, the target domain images are not “real” images but also generated with the preprocessing steps explained in section 4.1, applying certain appearance changes to the original (training) images using existing algorithms to simulate rainy or snowy conditions in the image lighting.
I find (as previously mentioned) an unclear point here: how are these “simulated” target domain images used to learn the natural variation model?  Both target domain images and the “augmented” training images during training (Natural variation model generated images) come from the same original labeled set.  I see a logical intuition behind it (if I apply certain augmentation (variation) to the training data, if the validation set contains images with similar variation applied, the performance will be better), but it is not a novel insight and current experimentation seems a bit biased if that’s the case.  If I understand the approach, the ideal situation to validate the proposed method would be to use real images from a target domain that does not have the semantic labels (maybe the Mapillary Vistas dataset could be a good source of data? it has data over different seasons and weather conditions), and then apply natural variations learned from this domain to the original labeled domain (labeled but with not such variations included).

Maybe a more detailed description of all this processing (maybe add some math expression?) would help to explain and distinguish these two cases, to present a more convincing validation set up:
- what’s the natural variation model applied and how is it obtained?
- what are the target domain images and how are they obtained?

**Summary Of The Paper:**

This work presents a new training algorithm for semantic segmentation deep neural networks (MRTAdapt) which is designed to obtain models more robust to changes in the scene due to natural variations, such as weather or lighting conditions.

The work poses this problem as a domain adaptation task, from the original domain (original labeled data) to the target domain (data with appearance changes due to lighting or weather).
The adaptation is achieved by training with augmented data simulating the natural variations learned from data. The training proposed combines a minimization of the segmentation loss (using the existing labels) and a maximisation of the probability of the generated images to be identified as images from the target domain, following GANs-like training, to ensure that the feature maps of the generated images are as similar as possible to target domain image feature maps.

In the experiments, the lighting and weather changes are simulated by the authors using existing tools applied on two public benchmarks. The approach is compared to other domain adaptation techniques obtaining higher accuracy in the segmentation of the simulated target domains.

**Summary Of The Review:**

I currently tend to reject. The problem tackled is relevant, and the intuitions behind the proposed approach interesting, however the technical novelty is limited and the experimental validation not sufficiently convincing on its current form as discussed in the weaknesses. Maybe authors can better discuss and explain the main concerns that hinder clarity on the novelty claims and the validation in the rebuttal.

---

> ### Author Response · Authors · 2021-11-18
> **Problem definition**
>
> We thank the reviewer for their comment and suggestion. The reviewer is right in that the problem of robust segmentation is modeled as a domain adaptation problem which is explicitly stated in the introduction of the revised manuscript. Also, to further explain why we mostly focus on domain adaptation methods in the literature review and in our evaluation, we would like to summarize the goal of this paper and the technical approach that we have considered.
> First, the objective of our paper is to train a semantic segmentation model that can achieve high accuracy even under out-of-distribution conditions such as natural variations (brightness or extreme weather conditions). Our work is motivated by the high cost of collecting and labeling data under different kinds of natural variation conditions. To mitigate this challenge, we develop a robust training algorithm with the help of domain adaptation techniques to capture the change of natural variations in advance.
> Our proposed training algorithm builds upon [Robey et al. 2020] and [Tsai et al. 2020]. In [Robey et al. 2020], a robust training algorithm is proposed for image classification tasks to enhance robustness against natural variation conditions. Note that the prediction target in image classification problems is usually of low dimension while the performance of that method is restricted by the performance of the natural variation model. To the contrary, in semantic segmentation problems, the prediction target is high-dimensional and complicated, the method proposed by [Robey et al. 2020]. might be not sufficient.
> We want to highlight that in our modified manuscript, we add the experiment data using the algorithm proposed by [Robey et al. 2020] for the Cityscapes experiment and the result shows that the existing algorithm that was proposed to solve the problem in image classification is not robust and scalable in semantic segmentation.
> To address this limitation, we developed a new training algorithm for semantic segmentation problem with the help of generative adversarial network (GAN) and the concept proposed by Tsai (AdaptSegNet) to minimize the gap on both image side and label side so that the performance is being improved simultaneously by domain adaptation techniques as well as GAN during the training time.
> We note that there are several existing works in the field of domain adaptation using labeled images from synthetic datasets to train the model that are robust to new real-world conditions. Thus, we believe that these methods provide a fair baseline to evaluate the effectiveness of our proposed method.

---

> ### Author Response · Authors · 2021-11-18
> **Related works**
>
> We thank the reviewer for their suggestion. To address this reviewer’s comments, in summary, we have made the following changes. First, we have rewritten the related work section for the domain adaptation with emphasis on the domain adaptation techniques that were applied in the field of semantic segmentation.
> We would also want to highlight that our method is aiming to improve the robustness of semantic segmentation under natural variation conditions with domain adaptation model being involved to capture natural variation. Besides, we have conducted comparative experiments shown in section 5 with several state-of-the-art domain adaptation techniques and showed that our proposed model outperformed other methods.
> In terms of image augmentation, we briefly introduced in the related work of robustness about the work proposed by [Hendrycks et al, 2020] that established rigorous benchmarks for image classification and proposed a series of image corruption operation methods. To be more specific, the natural variation representing the extreme weather case (snowy weather) being used in our paper is based on this corruption operation method.
> We would also note that in the modified manuscripts, we have expanded a bit more of our literature review on the image augmentation.

---

> ### Author Response · Authors · 2021-11-18
> **How the natural variation model and the target domain images are obtained**
>
> We thank the reviewer for their suggestion. Firstly, we would like to note that the objective of natural variation model is only to generate the corrupted images that have similar appearance as target-domain images from the source domain. Besides, we would like to highlight that in section 3.1, we mentioned that natural variation model can be customized regarding the needs of each dataset. In the section 4.2 (implementation details), we clarified that the natural variation model being used for Cityscapes dataset is powered by the multimodal unsupervised image-to-image translation method (MUNIT)proposed by Huang. The natural variation model being used in the Synthia dataset is powered by the RAIN (Random Adaptive Instance Normalization) model proposed by [Luo et al. 2020]. We would like to clarify that the training process of natural variation model is prior to the learning procedure of the segmentation model, so our proposed training algorithm mentioned in section 3 is not designed to learn the natural variation model. Also, we would like to highlight that the one of the major motivations of the existing works in the field of domain adaptation is that it is usually expensive and time-consuming to collect and label datasets in real-world setting. So, we discovered the majority of the datasets available online as open-source for training is collected in a relatively normal, non-perturbated environment such as sunny daytime so that the labeling procedures can be relatively easier. Our goal in this paper is trying to generate a model that could give high performance to abnormal conditions such as natural variations. To achieve this, we apply different natural variation models for different datasets to optimize the simulated effect for those natural variation conditions so that they have small gap with those real-world cases. For the obtaining process of the target domain images being used in our paper, we would like to highlight that in section 4.1 (data preprocessing), we change the HSV value of an image to different values so that an image would have different appearance in terms of brightness that look like real-world setting. For the snowing condition, we apply the corruption method proposed by Hendrick that was originally designed for ImageNet-C dataset.

---

> ### Author Response · Authors · 2021-11-18
> **Target domain data**
>
> We thank the reviewer for their detailed comments. We agree with the idea of this reviewer that there are some limitations of the validation set. For the reason that we do not have enough validation set that represent the target-domain images especially for the snowing case. In the previously submitted manuscript, we mentioned the HSV diving criteria for both datasets to serve as a basic partition of bright and dark images. We would like to emphasize that in our revised manuscript, we supplied another implementation detail that in order to enlarge the brightness effects, we manually changed the HSV value of the bright images by 80 HSV value for the Cityscapes so that the images that are labeled as “bright” will have stronger visual effects compared to the images labeled as “dark” in this dataset. Yet for the snowing case, for the reason that we do not have actual labeled images without any processing with snowing effects in both datasets, so to reduce the bias in this case, we blend images under all different kinds of brightness conditions to serve as source domain without snow effects and use the corrupted images after the snowing effect operation as target domain in this case.
> Meanwhile, we would like to note that in the previous work proposed by [Robey et al 2020] that focused on the robustness of image classification under natural variation also have similar evaluation process due to the lack of available datasets with similar natural variation effects.
> In the near future, we will keep working on more datasets with enough actual natural variations for training.

---

> ### Comment · Reviewer_ChGc · 2021-11-29
> **Comment after rebuttal feedback**
>
> I appreciate the authors' effort to provide explanations and additional details on their work, which will be taken into account for the final decision.

---

### Official Review · Reviewer_kE5M · 2021-11-02

**Correctness:** 3
**Technical Novelty And Significance:** 3
**Empirical Novelty And Significance:** 3
**Recommendation:** 6
**Confidence:** 4

**Main Review:**

In my opinion, the following are the strengths of the work:
 1) The use of the generator-discriminator (min-max problem) approach for tackling the natural variations in semantic segmentation is an interesting idea.
2) The focus on studying robustness with respect to natural variations (as compared to adversarial attacks) would be extremely useful in practice by eliminating the need for acquiring images in various environmental conditions for training the model.

However, there are a few major limitations of this work:
1) The proposed work utilizes two existing works (Robey et al 2020, and Tsai et al 2020) on domain adaptation for semantic segmentation. It will be better to include a summary of these works and clearly highlight how the proposed work is different from these works.
2) It is not clear how the natural variations are modeled in this work. At Page 5 (fourth para), it states the “natural variation model…… is learned from the data (Which data?)”. In section 4.1, the images are manually perturbed by varying the brightness or using ImageNet-C (for snow). I believe the way the natural variations are modeled plays an important factor in the robustness of the algorithm.
3) It is not clear what is the “Target Images” in this work? Does it refer to the perturbed images? If yes, why do only dark images are being considered as the target domain in Section 5 (Page 8, First paragraph).

In addition to the suggestions above, the following are some minor feedback:
4)In this work, all the models were trained with the same learning rate and the same number of epochs. I would suggest performing hyper-parameter tuning for all the models by varying the learning rate, weight decay, etc. This might result in a slight improvement in the performance.
5)It is interesting to note that FDA (Yang & Soatto 2020) performs poorly in the presence of snow, but there is no such effect observed in the dark images. If possible, it will be better to discuss this observation in more detail. This discussion might provide new insights into the performance of GANs.
6)It will be better to include few images and the corresponding labels obtained using the proposed approach. It will help the reader visualize the performance of the algorithm.
7) A minor suggestion: For better readability, it is recommended to arrange the existing methods in the same sequence in Tables 1 and 2.
8) Even though the experimental details are provided (including the hyperparameters), I would also recommend the authors publicly release the code to ensure reproducibility.


**Summary Of The Paper:**

 The paper proposes a discriminator-generator-based approach for robust semantic segmentation of images against natural variations such as snow, day/night (brightness). The input training image is first modified (perturbation in brightness, generating images with snow), and then fed to PSPNet to generate feature map for these perturbed images. Similarly, the original images (without perturbation) are fed to the same PSPNet to generate the original source feature map. The segmentation loss is a weighted sum of the loss computed on the original features and perturbed features. The loss on perturbed features finds the optimal perturbation parameter that maximizes the loss between predicted and original labels, while the loss on original image features is the traditional segmentation loss. The discriminator module attempts to distinguish between the features generated by perturbed images or target images. The proposed work is compared with the existing works on two standard datasets, Cityscapes and Synthia. The experimental results demonstrate the proposed approach outperforms the existing methods in terms of mIoU.  The main novelty of the proposed work is the combined use of adversarial networks and model-based training (Robey et al 2020) to achieve a robust semantic segmentation in the presence of variation in brightness and snow in the scene.


**Summary Of The Review:**

An interesting work proposing a semantic segmentation approach that is robust to variations in the lighting conditions (day/night) and the presence of snow in the scene.The main novelty of the proposed work is the combined use of adversarial networks and model-based training (Robey et al 2020) to achieve a robust semantic segmentation in the presence of variation in brightness and snow in the scene. However, a few important details about the approach is missing in the work. I would strongly recommend the authors address the limitations mentioned in Pt no 1,2 and 3. A clear definition/description of how the variations are modeled, what is the target images, and how work is different from the existing two works:  are needed before this work can be accepted for publication.

---

> ### Author Response · Authors · 2021-11-18
> **The highlight on the main contributions of our work**
>
> We thank the reviewer for their constructive comments and suggestions. In what follows we include detailed responses to this reviewer’s concerns.
> For limitation 1: We thank the reviewer for their suggestion. In the previously submitted manuscript, we briefly introduced the concept of the two existing works in the first section. To address this reviewer’s comment, in the revised manuscript we have expanded this discussion to further explain the contribution of our work with respect to these two existing works.
> In [Robey et al. 2020], an image classification-based robust training algorithm is proposed to enhance the robustness against natural variations. Specifically, a natural variation model is first obtained to capture the change in natural conditions (e.g., change in the lighting conditions). Then the training algorithm proposed in that work is applied at each iteration to find the worst-performed synthetic data generated by the natural variation model to serve as new data points for the model so that the trained model achieves high prediction accuracy under natural variations. Meanwhile, in [Tsai et al. 2020] the AdaptSegNet method is proposed based on the observation that images, that are seemingly different, may share similarities such as spatial layout and local context on feature map aspect. A discriminator network is proposed along with the original semantic segmentation network to minimize the distribution gap on label side aiming to help the model generate feature map outputs that look like the ones in target domain.
> Building on top of these two works, our proposed robust training algorithm, called MRTAdapt, aims to minimize the gap on both image and label side. To highlight the difference between our work and [Robey et al. 2020], recall that the latter is applied to image classification problems where the prediction targets are usually low-dimension and small in space. Meanwhile, the model-based robust algorithm proposed in [Robey et al. 2020] is highly dependent on the performance of the natural variation that serves to capture the change in natural variation conditions. These two issues make the algorithm less robust and less scalable in semantic segmentation task, where the prediction target is a high-dimension and complicated feature map. Inspired by the concept presented in [Tsai et al. 2020] that images might share similarities on label side even though their appearance is significantly different on the image side, we proposed a concept named semantic meaning invariance that the feature map of an image will remain unchanged regardless the change of natural variation conditions. In our proposed method, we apply the robust training algorithm with the help of natural variation model trying to expose the model with the worst performed synthetic target-domain images to enhance the robustness of image size. On the label side, we set up a discriminator to minimize the feature map gap between the output of synthetic target-domain images and true target-domain images to enhance the robustness on the label side. Compared to the method presented in [Tsai et al. 2020] that tries to minimize the gap of the outputs between source and target domain, our method is minimizing the gap between the outputs of synthetic ‘target’ domain and real target domain with the help of the natural variation model. We have also included extensive comparisons showing that our method outperforms related state-of-the-art works including [Tsai et al. 2020] as Table 1 and Table 2 shown in section 5 (experiment).

---

> ### Author Response · Authors · 2021-11-18
> **How the natural variations are modeled.**
>
> We thank the reviewer for their comment. In section 3.1, we highlighted that the natural variation model can be customized regarding the needs of each dataset. Depending on the number of semantic labels, which refers to the complexity of the prediction task, domain adaptation model might have different semantic segmentation performance on different datasets. We choose the best-fit domain adaptation model to capture the change of natural variation effects for the datasets we used to do experiments in this paper. In section 4.2 (implementation details), we mentioned that natural variation model is learned by the MUNIT (Multimodal Unsupervised Image-to-image Translation) proposed by [Huang et al] for the Cityscapes dataset, and the natural variation model used in the Synthia dataset is modeled by the RAIN (Random Adaptive Instance Normalization) model proposed by [Luo et al]. Applying a natural variation which can not capture the changes from source to target domain properly will cause the generated target domain image have huge appearance gap with the true target domain images and lead to a performance drop at inference time with the truth target domain images exposed to the model.

---

> ### Author Response · Authors · 2021-11-18
> **Target images**
>
> We thank the reviewer for their comment. The reviewer is indeed right that target images refer to perturbed images, i.e., images that are under natural variations. Specifically, in this paper, (i) source images refer to images in the normal case without being exposed to extreme appearance changes such as brightness or weather conditions and (ii) target images refer to images under natural variations (e.g., images under abnormal brightness condition or snow weather). As for the second point raised by this reviewer, the main reason why we treat images in daytime as source domain and dark images as target domain is because most datasets that are available online for semantic segmentation tasks have their images collected under sunny, normal-light and daytime condition to reduce the difficulty of model training and labeling. As mentioned in section 3.3, the major motivation of domain adaptation algorithms is the high cost of collecting and labeling images that come from new distributions. Thus, one of the objectives in our paper is to enhance the robustness of semantic segmentation models under those cases that are often not considered in existing datasets. This way, we can design models that can achieve high semantic segmentation performance in various environmental conditions even if those are not captured in training datasets. To address this reviewer’s comment, in the revised manuscript we define clearly the target images and we explain our choice for considering dark images as target images.

---

> ### Author Response · Authors · 2021-11-18
> **Modifications on the minor feedback**
>
> We thank the reviewer for their helpful feedback. To respond 4), we conducted multiple tests with different hyper-parameters. In the tables presented in our paper, the results were chosen using the best-performed cases among all the experiments we conducted.
> 5) FDA method proposed by Yang is based on the Fourier transform and its inverse, yet all the experiments being conducted in the paper only shows difference in terms of lightness of the image but not the snow condition.  6) In the revised manuscript, we append multiple visualization examples for different methods being tested in the paper under the appendix. 7) We would like to highlight that in the revised manuscript, we have made corresponding modifications to the tables using the suggestions provided by the reviewer. 8) We would also like to highlight that our code will be released soon for reproducibility.

---

### Decision · Program_Chairs · 2022-01-20

**Decision:**

Reject

**Comment:**

Overall, this paper receives negative reviews due to limited technical novelty and contributions. The reviewers discuss extensively on the merits of this work after the rebuttal phase. However, the authors' rebuttal does not address all the raised concerns. As such, the area chair agrees with the reviewers and does not recommend it be accepted at this conference